# Should the Treatment of Patients with Repeated Embryo Implantation Failure Be Adapted as a Function of the Endometrial Cytokine Profile? A Single-Center Experience

**DOI:** 10.3390/biomedicines11030817

**Published:** 2023-03-07

**Authors:** Bérangère Coutanceau, Esther Dos Santos, Nelly Swierkowski Blanchard, Anne Sanchez Louboutin, Florence Boitrelle, François Margueritte, François Vialard, Valérie Serazin, Khadija Fathallah

**Affiliations:** 1Department of Obstetrics and Gynaecology, Poissy-Saint-Germain-en-Laye Hospital, 78300 Poissy, France; 2Medical Biology Laboratory, Poissy-Saint-Germain-en-Laye Hospital, 78300 Poissy, France; 3RhUMA Team, UMR-BREED (INRAE, UVSQ, ENVA), UFR Simone Veil-Santé, 78180 Montigny le Bretonneux, France; 4Department of Anatomy and Pathology, Poissy-Saint-Germain-en-Laye Hospital, 78300 Poissy, France; 5Reproductive Biology Department, Poissy-Saint-Germain-en-Laye Hospital, 78300 Poissy, France; 6Department of Genetics, Poissy-Saint-Germain-en-Laye Hospital, 78300 Poissy, France

**Keywords:** embryo implantation, in vitro fertilization, endometrium, cytokine profile

## Abstract

Repeated embryo implantation failures (RIF) is a source of distress and frustration for patients and clinicians alike. Today’s approaches for treating RIF are largely empirical and have limited effectiveness. The main causes of RIF are poor endometrial receptivity and poor-quality embryos. Recent studies have suggested the involvement of immune dysregulation due to an imbalance between T-helper (Th) 1 and Th2 cytokines; this opens up perspectives for treating women with RIF and increasing the implantation rate. We conducted an interventional, longitudinal, prospective cohort study of the impact of correcting the cytokine imbalance on the clinical pregnancy rate in women with RIF. Seventy-seven women with RIF underwent an endometrial biopsy during the implantation window. The cytokine profile was evaluated by studying the activation and maturation of uterine natural killer (uNK) cells, the IL-15/Fn-14 mRNA ratio (a biomarker of uNK activation/maturation), and the IL-18/TWEAK mRNA ratio (a marker of angiogenesis and the Th1/Th2 balance). Personalized treatment was initiated for women with an abnormal endometrial cytokine profile (hyper-activation or hypo-activation). We documented the clinical pregnancy rate after subsequent embryo transfers. In total, 72.7% (56/77) of patients had an abnormal endometrial cytokine profile (hyper-activation in 68.8% (n = 53) and hypo-activation in 3.9% (n = 3). After treatment (or not) as a function of the endometrial profile, the overall clinical pregnancy rate was 30.2%. Our results indicated a potential positive effect of appropriate treatment on the ongoing pregnancy rate in women with RIF, despite the small number of cases analyzed. The results must now be validated in randomized studies with larger numbers of well-characterized patients. By applying a previously published decision tree, this treatment approach could be implemented in clinics worldwide.

## 1. Introduction

Despite recent progress in assisted reproductive technology (ART), today’s implantation rates per embryo transferred (20–30%), clinical pregnancy rates per transfer, and live birth rates are unsatisfactory [1]. Repeated implantation failure (RIF) is defined as the absence of an ongoing clinical pregnancy after the transfer of at least four good-quality embryos [2]. The management of RIF remains a challenge for ART professionals.

A poor-quality embryo is the most important cause of RIF. Many studies have confirmed the negative impact of embryonic aneuploidies on the likelihood of pregnancy [3]. Although a recent study showed that the routine pre-implantation diagnosis of aneuploidy (PGD-A) enabled pregnancy to be achieved after at least three attempts in more than 90% of cases [4], RIF can sometimes occur even when good-quality embryos are transferred. In fact, the literature data indicate clearly that in humans fewer than 60% of euploid embryos result in pregnancy [5].

Along with well-known causes of embryo implantation failure (such as anatomical anomalies [6], infectious diseases [7], hormone disorders [8], hematological disorders [9], and immune diseases [10]), it appears that endometrial defects constitute a non-negligible risk factor for RIF [11]. For example, it is well known that immune defects can cause obstetric disorders in early pregnancy and during placental development [12]. Defects in maternal-fetal immune tolerance are thought to be strongly associated with morbidities such as pregnancy-induced hypertension, pre-eclampsia, growth retardation, and preterm delivery [13].

Embryo implantation is a complex process during which the embryo adheres to and then penetrates the maternal endometrium. Various adhesion molecules, growth factors, chemokines, and cytokines create a molecular dialogue between the embryo and the endometrium. Two groups of cytokines have different roles in the polarization of T cell secretion: Th1 type cytokines (interleukin 2 (IL2), tumor necrosis factor beta (TNFß), and interferon gamma (IFNγ)) are involved in cell-mediated immune reactions that are harmful for implantation, whereas Th2 type cytokines (IL4, IL5, and CD40) are involved in humoral-mediated immune reactions that favor implantation. Furthermore, uterine-specific natural killer (uNK) cells have a different phenotype from other circulating NK cells with a low level of spontaneous cytotoxicity that can be modulated by the cytokine environment [14] and by class 1b major histocompatibility complex molecules (including HLA-G). Overall, the establishment of an equilibrium enables implantation and favors an ongoing pregnancy. Any imbalance will result in implantation failure and rejection of the fetal allograft [15]. For example, an abnormally high basal level of uNK cell activity is reportedly a predictor of recurrent miscarriages [16].

Following on from these observations, several studies have shown that the expression of IL15 and IL-18 is dysregulated in the endometrium of women with RIF, relative to that of fertile women [17]. IL-15 activity is modulated by its receptor fibroblast growth factor-inducible molecule 14 (Fn-14) [18] and is directly involved in the recruitment and maturation of uNK cells [19]. Similarly, the Th2-type cytokine IL-18 is immunomodulated by the TNF weak inducer of apoptosis (TWEAK) [20]. The expression of IL-18 rises during the implantation window and the cytokine has a primarily angiogenic role [21]. By monitoring the IL-15/Fn-14 mRNA ratio (a biomarker of uNK activation/maturation) and the IL-18/TWEAK mRNA ratio (a biomarker of angiogenesis and Th1/Th2 balance), the human endometrium’s immune profile can thus be classified as normal, hyper-activated, or hypo-activated [22].

In the context of immune deregulation and RIF, it was first shown that the prednisone treatment of women with a hyper-activated profile led to the normalization of immune biomarker levels [23]. This normalization was associated with a live birth rate of 39.8% after the first embryo transfer [22]. We therefore hypothesized that the correction of an abnormal endometrial immune profile would result in a higher pregnancy rate.

The primary objective of the present study was to describe the management of RIF and the clinical pregnancy rates in our center as a function of the endometrial immune profiles. The secondary objective was to evaluate the impact of the time interval between the diagnosis of endometrial immune dysfunction (following an endometrial biopsy) and the implementation of the ART cycle.

## 2. Materials and Methods

We performed an interventional, longitudinal, prospective cohort study at the Poissy-Saint-Germain-en-Laye Hospital over a four-year period (from January 2016 to December 2019). All women with RIF were invited to undergo an endometrial biopsy for an analysis of their cytokine profile. The treatment prior to embryo transfer was then adjusted as a function of the profile (normal, hyper-activated, or hypo-activated). In each group, the ongoing pregnancy rate per transfer cycle was calculated for the 12 months following the biopsy.

We also analyzed the miscarriage rate; the ongoing pregnancy rate per treated patient; the mean time interval between the endometrial biopsy and a pregnancy; the number of patients who received treatment after a biopsy; the number of cycles with a transfer (with one or two embryos); the number of embryos transferred in total and on average per patient after a biopsy; and the mean time interval between the biopsy and the embryo transfer.

### 2.1. Population

In the absence of a consensus of the definition in the literature, we defined RIF as the absence of an ongoing pregnancy at more than 10 weeks of amenorrhea despite embryo transfer at the 2 or 3-day culture stage or four embryos at the blastocyst stage during an in vitro fertilization (IVF)/intracytoplasmic sperm injection (ICSI) procedure. All included patients had to be between 18 and 43 years of age. Patients outside this age range; those having undergone a gamete donation procedure; those who had not undergone an endometrial biopsy during hormone replacement therapy and during the implantation phase (according to the histologic features); and those wishing to stop their treatment were excluded. All included patients gave their written consent to participate in this study.

### 2.2. Endometrial Biopsy

An endometrial biopsy (using a Cornier endometrial suction catheter) was performed in all patients with a spontaneous menstrual cycle during the implant phase (between days 20 and 24 of the cycle, for normally ovulating patients) without anesthesia but with ultrasound guidance, if necessary. The aspirated fragments of uterine mucosa were aliquoted into two different vials: (i) a 4% formalin solution for standard histological analysis, in order to confirm that the specimen was in the secretory phase, and (ii) RNA stabilization solution (RNAlater; Qiagen, Venlo, The Netherlands), to study the expression of the genes of interest.

### 2.3. RNA Quantification

Total RNA from the cells was extracted with the Nucleospin RNA II^®^ kit (Macherey-Nagel, Düren, Germany) and stored at −80 °C. The RNA was reverse-transcribed using the SuperScript III Reverse Transcriptase kit (Invitrogen, Waltham, MA, USA) and the expression levels of the target gene (coding for IL15, IL18, Fn14, and TWEAK) were quantified with regard to the reference genes (coding for beta 2 microglobulin, ribosomal protein L13A, and TATA box-binding protein). The primer sequences have been described previously [21].

For amplification, the reaction mixture consisted of 4 µL of cDNA diluted 1:20, 0.2 µL of forward primer (25 µm), 0.2 µL of reverse primer (25 µm), 0.6 µL of sterile water, and 5 µL of buffer (2×) containing Sybr^®^Green, Taq polymerase, and its buffer (Light Cycler 480 SYBR Green I Master mix, Roche Diagnostics, Basel, Switzerland). Quantitative RT-PCR was performed on a Light Cycler 480 (Roche Diagnostics). The program used included a 7 min denaturation initiation at 95 °C (to activate the Taq polymerase), 44 PCR cycles (10 s of denaturation at 95 °C, the hybridization of the coupled primers at 60 °C for 10 s, and then elongation by Taq polymerase for 15 s), and a melting step (65–95 °C, with five acquisitions/s) and a cooling step at 4 °C. Each run included a cDNA-free control and an inter-run calibrator. The results were interpreted as relative amounts using Light Cycler 480 software 1.5.0 (Roche Diagnostics), as described previously [21,22].

We calculated the mean ± standard deviation (SD) IL15/Fn14 and IL18/TWEAK mRNA ratios in endometrial biopsies in a control group of fertile women. For each patient, the mRNA ratio was considered to be low when it was more than one SD below the mean. Conversely, the ratio was considered to be high when it was more than one SD above the mean.

### 2.4. Immunohistochemistry of uNK Cells

Biopsies were fixed in formalin for 24 h and embedded in kerosene. Histologic sections were stored at 4 °C before use. Labeling was automated with the BENCHMARK XT Ventana system, using the Ultra View Universal DAB Kit and the CD56/NCAM clone 123 C3 Ventana antibody (Abbott, Chicago, IL, USA). The method used to interpret the results has been described previously [23]. The uNK cell count was calculated as the mean (per field) number of CD56+ cells in four representative fields at a magnification of 400×, on a microscope with a field diameter of 0.62. A uNK cell count of 30 to 60 was considered to be normal; hence, an abnormally low count was <30, and an abnormally high count was >60.

### 2.5. Interpretation of the Cytokine Profile

The cytokine profile was classified as normal, hyper-activation, or hypo-activation according to a previously described decision tree [22] (Figure 1) [24]. To classify the patients, we sequentially used the IL18/TW ratio, the uNK cell count (when the IL18/TW ratio was normal), and the IL15/Fn14 ratio (when the uNK cell count was normal or high). 

### 2.6. Treatment Regimens in Each Group

#### 2.6.1. Treatment in the Hyper-Activation Group

The treatment goal was to control the activation of immune cells, promote the invasion by the embryo, and avoid the rejection and apoptosis of the embryo. Our treatment strategy was based on the fact that (i) glucocorticoids have long been used to influence the expression and activity of pro-inflammatory mediators and are key components of many immunomodulatory regimens [25,26] and (ii) progesterone and estrogen decrease the expression of pro-inflammatory cytokines [27].

We therefore applied a previously described treatment protocol [22] with the administration of prednisolone (20 mg per day) and vitamin E (Toco^®^ 500 mg, twice a day) from day 3 of the cycle onwards. The corticosteroid treatment was continued until 8 weeks after the embryo transfer and was then tapered over a 3-week period if the pregnancy test was positive. Treatment with vitamin E was continued until 10 weeks after transfer. High-dose progesterone supplementation was also initiated (Progestan^®^ 200 mg, two tablets for intravaginal administration three times a day) on the evening of the oocyte retrieval (for a fresh embryo transfer) or when the uterine mucosa during estrogen treatment was thicker than 7 mm (for a frozen embryo transfer during hormone replacement therapy). Oral estrogen treatment (Provames^®^ 2 mg, two tablets per day) was initiated on the evening of the oocyte retrieval (for a fresh embryo transfer) or on the first day of the cycle (for a frozen embryo transfer during hormone replacement therapy). These treatments were continued for 10 weeks if the pregnancy test was positive. The patient was instructed to avoid activities that might damage the endometrium during the cycle preceding the transfer, and sexual intercourse was prohibited after the transfer. In fact, seminal plasma has been shown to induce the mobilization and activation of local immune cells [28]. Lastly, it has been reported that endometrial scratching or other endometrial injury in the middle of the luteal phase of the cycle preceding the embryo transfer activates and stimulates the expression of adhesion molecules and IL-15 [29]. Human chorionic gonadotropin (HCG) triggers the proliferation and maturation of uNK cells and induces immune tolerance at the maternal-fetal interface [30].

#### 2.6.2. Treatment in the Hypo-Activated Group

The treatment goal was to increase local responsiveness and promote the adhesion of the embryo. Endometrial scratching was performed in the middle of the luteal phase of the cycle preceding the embryo transfer. Luteal phase support with HCG 1500 IU was provided on days 4, 6 and 8 after oocyte retrieval. Sexual intercourse after the transfer was recommended.

#### 2.6.3. Treatment in the Normal-Profile Group

The patients in the normal-profile group did not receive any additional treatments.

### 2.7. Statistical Analysis

First, we analyzed the following characteristics in the three endometrial profile groups (normal, hyper- and hypo-activation group): age, body mass index, tobacco use, the serum anti-Mullerian hormone level, the presence or absence of endometriosis, the type of infertility, parity, gravida, history of miscarriage, rank of the attempt, and the mean number of embryos transferred before the biopsy. 

Second, we assessed the number of cycles with a transfer after the biopsy and also the number of embryos transferred after the biopsy for treatment into each group. Moreover, a treated patient could have more than one cycle after the biopsy (depends of the number of attempts) and, within each cycle, more than one embryo transferred. Within each group of the endometrial profile, we assessed the mean number of embryos transferred, the mean time interval between the biopsy and the transfer, and the number of pregnancies and ongoing pregnancies between each cycle with a transfer. 

Quantitative variables are expressed as the mean standard deviation, minimum, and maximum, and were compared by the status of the endometrial profile using non-parametric Kruskal-Wallis test. Because of the distribution between groups (such as the hypo group), assessment of normality cannot be performed and non-parametric tests were used. Qualitative variables were compared by endometrial profiles using Fisher exact tests.

The significance threshold retained for all analyses (p) was an alpha risk of 0.05. All analyses were also performed using the STATA© 15.1 IC (StataCorp, College Station, TX, USA).

## 3. Results

Between January 2016 and December 2019, 120 patients underwent an endometrial biopsy and a cytokine profile analysis of an indication of RIF. This study’s patient flow is summarized in Figure 2. Twenty-one of the 120 patients (17.5%) were subsequently excluded because the endometrium was not in the implantation phase (judging by the histologic features) or the patient had transferred to a gamete donation program. Of the remaining 99 biopsies, 22 (22.2%) did not yield a result (due to a lack of material (in 90.5% of these cases)) or the presence of degraded RNA (in 9.5%C)., whereas 77 (77.8%) yielded a result.

Hence, a total of 77 patients were included in the final analysis (Table 1). The mean (range) age of the patients was 34.2 years (24–42). The mean number of embryos transferred before the endometrial biopsy was 7.5. Twenty-two of the patients had experienced at least one early miscarriage during their ART program prior to the biopsy (29%). We did not observe any significant differences between the three groups for any of the variables (*p* > 0.05), except for the AMH serum level, which is the highest in the normal group compared to the others

Fifteen of the 21 patients (71.4%) in the “normal profile” group underwent at least one embryo transfer (fresh or frozen) within 12 months of the endometrial biopsy, with an average of 3.5 embryos transferred per patient treated and a mean time interval between the biopsy and the transfer of 5.1 months (Table 2). Seven ongoing pregnancies were obtained in 33 cycles with a transfer (i.e., a rate of 21.2%). The mean time interval between the biopsy and the pregnancy was 6.3 months.

After the implementation of appropriate treatment, 30 of the 53 patients in the “hyper-activation” group (56.6%) underwent an embryo transfer within 12 months of the biopsy. On average, 2.8 embryos were transferred per patient treated, and the mean time interval between the biopsy and the transfer was 6.1 months (Table 2). Fifteen ongoing pregnancies were obtained in 52 cycles with a transfer (i.e., a rate of 28.8). The mean time interval between the biopsy and the pregnancy was 6.8 months.

In the “hypo-activation” group, two of the three patients (66.7%) underwent an embryo transfer after treatment. In five cycles, five embryos were transferred, and one live birth was obtained. The mean time interval between the biopsy and the transfer was 3.8 months for the latter pregnancy (Table 2).

We did not observe any significant differences between the three groups for treated patients within any of the variables (mean number of embryos transferred, mean interval time between a biopsy and a transfer, number of pregnancies). 

## 4. Discussion

Repeated implantation failure is one of the factors that limits the success of ART. Many lines of research are underway in this field, with a view of increasing the pregnancy rate. In fact, RIF is probably a multifactorial phenomenon. Historically, RIF was considered to have an anatomical cause, such as a congenital or acquired anomaly of the cavity. Today’s biological and immunologic approach appears to be opening up interesting new perspectives [31], particularly with regard to the endometrium’s immune profile.

In the present study, we sought to duplicate research performed in 2016 [22] in order to determine whether adapting the treatment of patients with RIF as a function of the endometrial cytokine profile could increase pregnancy rates. We adopted the same definition of RIF as used previously, namely the absence of pregnancy for more than 10 weeks of amenorrhea after the transfer of at least six embryos during a course of IVF/ICSI. Likewise, we used the same algorithm and the same thresholds for the interpretation of the results. The mean number of embryos transferred in our study population of RIF patients was 7.5; this value is similar to those reported in the literature [32,33]. However, our interventional, longitudinal, prospective cohort study with the same methodology as the previously reported study also had the same limitations. To validate the treatment strategy, a randomized controlled trial would be required.

As mentioned above, a high proportion (22.2%) of the histologically valid biopsies did not yield a result for the cytokine profile—due mainly to a lack of endometrial material or the presence of degraded RNA. These data emphasize the operator-dependent nature of the endometrial biopsy.

The proportion of abnormal endometrial profiles in our cohort (72.7%) was similar to that reported by Lédée et al. (80%) [22]. Conversely, we found a higher proportion of hyper-activation profiles (68.8%, versus 56% for Lédée et al.) and thus a lower proportion of hypo-activation profiles (3.9% versus 25%, respectively). These differences might be due to the small number of patients in our study (120, versus 394 in Lédée et al.’s study).

In our study, the ongoing pregnancy per patient treated rate was 46.7% in the “normal profile” group; despite the presence of RIF, this value is higher than that found in the general population of infertile patients. This relative improvement might be due to an effect of the biopsy or “scratching” on implantation, as often observed in the literature [34]. This effect might be mediated by the local inflammation created by the invasive procedure [35].

The rate for an ongoing pregnancy per cycle with a transfer (cumulative transfers in the 12 months after the biopsy) was 28% in the hyper-activation and hypo-activation groups together (with an average of 2.8 embryos transferred per patient treated after the biopsy). This value is similar to the 24.2% observed in the general population of infertile women after IVF/ICSI with frozen embryo transfers [36]. Hence, personalized treatment as a function of the endometrial profile might therefore “normalize” the likelihood of im-plantation in a population with RIF.

Nevertheless, this ongoing pregnancy rate is lower than the 40.9% reported in our original study [22]. This difference might be due to the severity of RIF in the present study population, as again evidenced by the average of 7.5 fruitless embryo transfers before the biopsy. However, our results are similar to those in two other studies, notably for the “hyper-activation” group treated with prednisone [23,25].

In the hyper-activation and hypo-activation groups, the pregnancy rate did not differ significantly from that observed in the control group. This similitude indicates that normalization of the immune profile resulted in a “normal” pregnancy rate, although we did not check for normalization in a subsequent endometrial biopsy. In order to confirm this result, one would have to conduct a randomized study in which a group of patients with hyper-activation or hypo-activation is treated for deregulation and another group is not. Nevertheless, given our encouraging results and the great distress of couples faced with RIF, we consider that this option was not ethically feasible. In order to generate biological proof of our treatment’s putative ability to normalize the cytokine profile, we could have performed post-treatment biopsies. However, previous studies have demonstrated the beneficial impact of corticosteroids on the normalization of immune biomarkers for immune hyper-activation profiles [23] and that an endometrial biopsy activates endometrial immune cells [37].

With regard to the time interval between biopsy and embryo transfer, the vast majority of published studies focused on the cycle preceding the biopsy [38]. However, it has been shown [29] that monocytes recruited at the injured sites (which have a role in improving endometrial receptivity) are long lived and are found in the tissues for several months [39,40]; this might mediate the biopsy’s long-term effect. In our study, the mean time interval between the biopsy to pregnancy was 6.3 and 6.8 months in both the “normal” and “hyper-activation” groups, respectively, and 3.8 months in the “hypo-activation” group. This implies that a short time interval between the biopsy and transfer is not essential for a successful attempt. One could therefore consider not repeating the biopsy in the following 12 months, while continuing to transfer embryos and while applying the recommended treatment scheme as a function of the cytokine profile. However, this approach would also need to be validated in randomized trials with follow-up biopsies at different time intervals.

The main limitation of our study was the small number of participants, given the low incidence of RIF in ART (estimated to be around 10%) [41]. However, the lack of a consensus of the definition of RIF makes this proportion difficult to estimate. Secondly, a high proportion of the endometrial biopsies did not give an interpretable result due to a lack of material or the absence of an implantation window. Thirdly, we did not calculate the number of participants required prospectively. Fourthly, we did not use diagnose aneuploidies prior to implantation as this technique is still controversial but has been shown to be very effective in increasing pregnancy rates [4]. Lastly, and as mentioned above, this was only an interventional, longitudinal, prospective cohort study that lacked a control group. However, it is very difficult not to suggest a new strategy for couples with RIF.

In terms of study strengths, our population was very homogeneous (despite its small size) and produced results that were consistent with the literature data [22]. Our study was also the first to have assessed ongoing pregnancy rates over a long period (12 months) after the endometrial biopsy; most of the previous studies focused solely on the cycle immediately after the biopsy [38,42]. However, this is a pilot study to assess the performance of this strategy. In order to confirm these results, a prospective study based on these findings and with a number of subjects of “needed to treat” is necessary. Then, a prospective randomized study should be performed to confirm the efficacy of this clinical approach, aiming to improve implantation rates per embryo transferred.

## 5. Conclusions

Our results indicated a potential positive effect of appropriate treatment on the ongoing pregnancy rate in women with RIF. The results must now be validated in randomized studies with a larger number of well-characterized patients. By applying a previously published decision tree, this treatment approach could be implemented in clinics worldwide.

## Figures and Tables

**Figure 1 biomedicines-11-00817-f001:**
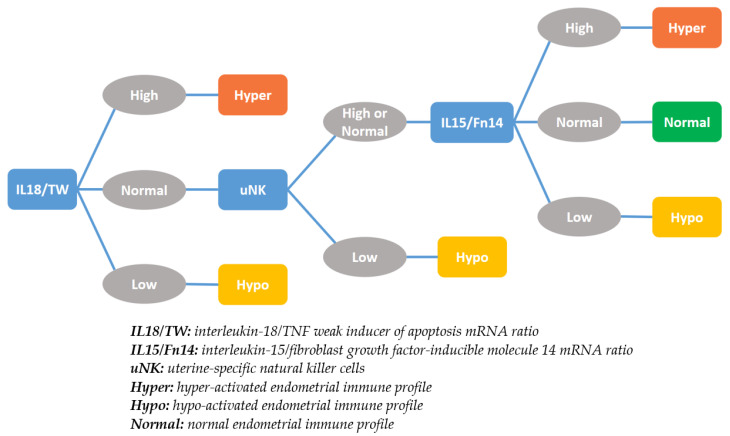
Decision tree for the interpretation of immune deregulation, according to Lédée et al. [24].

**Figure 2 biomedicines-11-00817-f002:**
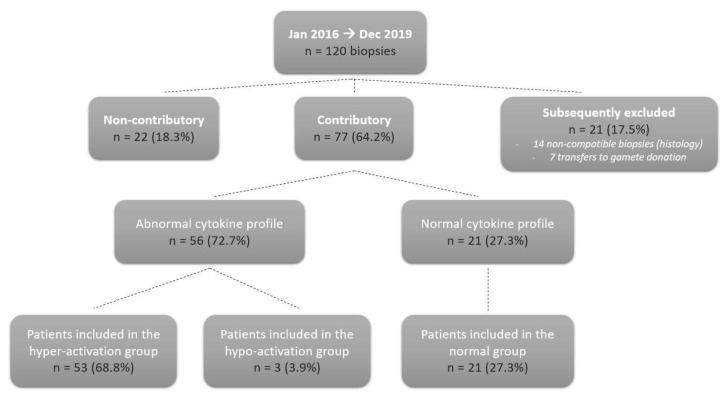
Study flow diagram.

**Table 1 biomedicines-11-00817-t001:** The characteristics of the study participants, according to their endometrial profile.

Endometrial Profile	Normal Group 21/77 (27.3%) n (%) or µ ± SD	Hyper-Activation Group 53/77 (68.8%) n (%) or µ ± SD	Hypo-Activation Group 3/77 (3.9%) n (%) or µ ± SD	*p* Value
**Age (years) (n = 77)**	34 (25–42)	34 (24–41)	32 (30–33)	0.27
**Body mass index (kg/m^2^) (n = 76)**	23 (18–31)	24 (17–34)	22 (19–25)	0.58
**Tobacco use (n = 74)**	4 (19.1%)	6 (12.0%)	0 (0.0%)	0.66
**Serum AMH level (ng/dL) (n = 69)**	4.18 (0.7–10.6)	2.94 (0.1–14.7)	3.37 (2.0–4.1)	0.04
**Endometriosis (n = 77)**	7 (33.3%)	14 (26.4%)	1 (33.3%)	0.81
**Primary infertility (n = 77)**	14 (66.7%)	33 (62.3%)	3 (100.0%)	0.62
**Gravida (n = 77)**				0.86
**G0**	7 (33.3)	23 (43.4)	2 (66.7)	
**G1**	8 (38.1)	17 (32.1)	1 (33.3)	
**G2+**	6 (28.6)	13 (24.5)	0(0.0)	
**Parity (n = 77)**				0.45
**P0**	15 (71.4)	41 (77.4)	3 (100.0)	
**P1**	6 (28.6)	8 (15.1)	0 (0.0)	
**P2+**	0 (0.0)	4 (7.6)	0 (0.0)	
**At least one miscarriage before the ART program (n = 77)**	11 (52.4%)	21 (39.6%)	1 (33.3%)	0.63
**At least one miscarriage during the ART program (n = 77)**	8 (38.1%)	13 (24.5%)	1 (33.3%)	0.46
**Mean Attempted rank (n = 77)**	2.4 ± 0.8	2.5 ± 0.8	2.3 ± 0.6	0.88
**Mean Embryos transferred before the biopsy (n = 77)**	7.8 ± 2.3	7.5 ± 2.1	6.3 ± 0.6	0.48

AMH: anti-Mullerian hormone; ART: assisted reproductive technology.

**Table 2 biomedicines-11-00817-t002:** Cumulative results for transfer within 12 months for treated patients.

	Normal Group n = 15 n (%) or µ ± SD	Hyper Group n = 30 n (%) or µ ± SD	Hypo Group n = 2 n (%) or µ ± SD	*p* Value
**Cycles with a transfer after the** **biopsy for treated patient (n)**	33	52	5	
**Embryos transferred after the** **biopsy for treated patient (n)**	52	84	5	
**Mean number of embryos transferred per treated patient, after the biopsy ***	3.5 ± 2.3	2.8 ± 1.6	2.5 ± 2.1	0.61
**Mean time interval between the biopsy and the transfer (months) ****	5.1 ± 3.4	6.1 ± 3.4	4.5 ± 0.7	0.48
**Pregnancies per cycle patients with a transfer ****	10/33 (30.3%)	19/52 (36.5%)	1/5 (20.0%)	0.70
**Ongoing pregnancies per cycle with a transfer ****	7/33 (21.2%)	15/52 (28.8%)	1/5 (20.0%)	0.84
**Miscarriages per cycle with a transfer ****	3/33 (9.1%)	3/52 (5.8%)	0/5 (0.0%)	0.77
**Ongoing pregnancies per treated patient ***	7/15 (46.7%)	13/30 (43.3%)	1/2 (50.0%)	1.00
**Mean ± SD time interval between the biopsy and the pregnancy (months) *****	6.3 ± 3.2	6.8 ± 3.5	3.8 ± NA	0.59

*: per treated patients: normal group n = 15, hyper group n = 30, hypo group n = 2; **: per cycle per treated patients: normal group n = 33, hyper group n = 52, hypo group n = 5; ***: per pregnancy: normal group n = 10, hyper group n = 19, hypo group n = 1; NA: Not Available, only one patient.

## Data Availability

The data presented in this study are available from the corresponding author upon reasonable request.

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
