# Peer review of "Should the Treatment of Patients with Repeated Embryo Implantation Failure Be Adapted as a Function of the Endometrial Cytokine Profile? A Single-Center Experience"

_biomedicines, 2023, doi:10.3390/biomedicines11030817_

Round 1

Reviewer 1 Report (Previous Reviewer 3)

The authors have adequately made all suggested changes.

Author Response

Dear editors,

Firstly, my co-authors and I would like to thank you again, and the reviewers, for giving us the opportunity to improve our manuscript, in order to publish it into biomedicines.

Reviewer 1:

The authors have adequately made all suggested changes.

Thank you

Reviewer 2 :

Regarding this study, there are some questions that the authors should answer:

A very important limitation of the study is the small number of cases analyzed (77 patients), of which only 47 were treated and 15 of the normal group. These data should appear in the Abstract section.

Answer : It has been done in the abstract, line 41: “despite the small number of cases analyzed.”

With these patients it is difficult to reach conclusions. Why is a calculation of the necessary sample size not made to reach conclusions about the objectives set?

Answer : As answered earlier, this is a pilot study describing what is actually done to improve the success rate in case of hyper or hypoactivation. So, a sample calculation was not done and is not mandatory in this case. Of course, in another study, to confirm these results, a "needed to treat" number will be performed based on our data.

A sentence has been added to the discussion section, line 392-393: “In order to confirm these results, a prospective study based on these findings and with a number of subjects of “needed to treat” is necessary.”

The authors indicate that they used the non-parametric Kruskal-Wallis test. However, they do not previously expose how the normality or otherwise of the variables studied was assessed. In the variables that are normal, the 3-group comparison test should be ANOVA. The authors should review whether the results in Table 2 are correct, taking into account the statistical test used according to the normality of the variables of the 3 groups.

Answer : Given the small number of patients within the three groups, and such for the hypo group, assessment of normality couldn’t be performed or is not adapted in this case. This is the reason why a non parametric test has been used as requested previously by another reviewer and validated. But, and according to this remark, a sentence has been added in the method section, line 251-252: “Because of the distribution between groups (such as the hypo group), assessment of normality cannot be performed, and non parametric tests were used”

Table 1 and 2. The explanations of the variables should be at the bottom of the table.

Answer : It has been done accordingly

Table 1. The mean values should be better accompanied by their dispersion (± SD).

Answer : It has been done accordingly

Table 2. What does 3.8 ± NA mean?

Answer : Not Avaible because there only one women in this group, so SD couldn’t be calculated. This has been added to the legend.

In the results, the authors find that there were no significant differences between the normal (untreated) group and the hyper and hypo groups, but we do not know what the untreated results of the abnormal groups would be. For this, a randomized prospective clinical trial would be required. This should be dealt with more fully in the Discussion section and assessed with reference to the existing literature.

Answer : This has been added and discussed line 415 to 419 : “However, this is a pilot study to assess the performance of this strategy. In order to confirm these results, a prospective study based on theses findings and with a number of subjects of “needed to treat” is necessary. Then, a prospective randomized study should be performed to confirm the efficacy of this clinical approach, aiming to improve implantation rates per embryo transferred.”

Academic editors: Update materials

Answer : This has been modified and updated following reviewers expectations.

English language and style are fine/minor spell check required:

Answer : According to Isabelle Cheng, assistant editor, English service will be offer after paper acceptance.

Reviewer 2 Report (Previous Reviewer 1)

Manuscript ID: biomedicines-2224959

Title: Should the treatment of patients with repeated embryo implantation failure be adapted as a function of the endometrial cytokine profile? A single-center experience.

Coutanceau et al. present a study in which” the main objective was to describe the management of the repeated embryo implantation failures (RIF) and clinical pregnancy rates in a center based on endometrial immunity profiles. The secondary objective was to assess the impact of the time interval between the diagnosis of endometrial immune dysfunction (after endometrial biopsy) and the implementation of the assisted reproductive technology (ART) cycle”.

 Regarding this study, there are some questions that the authors should answer:

 A very important limitation of the study is the small number of cases analyzed (77 patients), of which only 47 were treated and 15 of the normal group. These data should appear in the Abstract section.

 With these patients it is difficult to reach conclusions. Why is a calculation of the necessary sample size not made to reach conclusions about the objectives set?

The authors indicate that they used the non-parametric Kruskal-Wallis test. However, they do not previously expose how the normality or otherwise of the variables studied was assessed. In the variables that are normal, the 3-group comparison test should be ANOVA. The authors should review whether the results in Table 2 are correct, taking into account the statistical test used according to the normality of the variables of the 3 groups.

 Table 1 and 2. The explanations of the variables should be at the bottom of the table.

 Table 1. The mean values ​​should be better accompanied by their dispersion (± SD).

 Table 2. What does 3.8 ± NA mean?

In the results, the authors find that there were no significant differences between the normal (untreated) group and the hyper and hypo groups, but we do not know what the untreated results of the abnormal groups would be. For this, a randomized prospective clinical trial would be required. This should be dealt with more fully in the Discussion section and assessed with reference to the existing literature.

Author Response

Dear editors,

Firstly, my co-authors and I would like to thank you again, and the reviewers, for giving us the opportunity to improve our manuscript, in order to publish it into biomedicines.

Reviewer 1:

The authors have adequately made all suggested changes.

Thank you

Reviewer 2 :

Regarding this study, there are some questions that the authors should answer:

A very important limitation of the study is the small number of cases analyzed (77 patients), of which only 47 were treated and 15 of the normal group. These data should appear in the Abstract section.

Answer : It has been done in the abstract, line 41: “despite the small number of cases analyzed.”

With these patients it is difficult to reach conclusions. Why is a calculation of the necessary sample size not made to reach conclusions about the objectives set?

Answer : As answered earlier, this is a pilot study describing what is actually done to improve the success rate in case of hyper or hypoactivation. So, a sample calculation was not done and is not mandatory in this case. Of course, in another study, to confirm these results, a "needed to treat" number will be performed based on our data.

A sentence has been added to the discussion section, line 392-393: “In order to confirm these results, a prospective study based on these findings and with a number of subjects of “needed to treat” is necessary.”

The authors indicate that they used the non-parametric Kruskal-Wallis test. However, they do not previously expose how the normality or otherwise of the variables studied was assessed. In the variables that are normal, the 3-group comparison test should be ANOVA. The authors should review whether the results in Table 2 are correct, taking into account the statistical test used according to the normality of the variables of the 3 groups.

Answer : Given the small number of patients within the three groups, and such for the hypo group, assessment of normality couldn’t be performed or is not adapted in this case. This is the reason why a non parametric test has been used as requested previously by another reviewer and validated. But, and according to this remark, a sentence has been added in the method section, line 251-252: “Because of the distribution between groups (such as the hypo group), assessment of normality cannot be performed, and non parametric tests were used”

Table 1 and 2. The explanations of the variables should be at the bottom of the table.

Answer : It has been done accordingly

Table 1. The mean values should be better accompanied by their dispersion (± SD).

Answer : It has been done accordingly

Table 2. What does 3.8 ± NA mean?

Answer : Not Avaible because there only one women in this group, so SD couldn’t be calculated. This has been added to the legend.

In the results, the authors find that there were no significant differences between the normal (untreated) group and the hyper and hypo groups, but we do not know what the untreated results of the abnormal groups would be. For this, a randomized prospective clinical trial would be required. This should be dealt with more fully in the Discussion section and assessed with reference to the existing literature.

Answer : This has been added and discussed line 415 to 419 : “However, this is a pilot study to assess the performance of this strategy. In order to confirm these results, a prospective study based on theses findings and with a number of subjects of “needed to treat” is necessary. Then, a prospective randomized study should be performed to confirm the efficacy of this clinical approach, aiming to improve implantation rates per embryo transferred.”

Academic editors: Update materials

Answer : This has been modified and updated following reviewers expectations.

English language and style are fine/minor spell check required:

Answer : According to Isabelle Cheng, assistant editor, English service will be offer after paper acceptance.

Round 2

Reviewer 2 Report (Previous Reviewer 1)

Authors make changes to the manuscript that improve it.

This manuscript is a resubmission of an earlier submission. The following is a list of the peer review reports and author responses from that submission.

Round 1

Reviewer 1 Report

Journal: Biomedicines

Manuscript ID: biomedicines-2016683

Title: Could therapeutic adaptation according to the endometrial cytokine profile in patients with repeated embryo implantation failure be usefull: a monocentric experience.

The aim of this study was to evaluate the impact of correction of cytokine imbalance on clinical pregnancy rate in 77 patients with repeated implantation failure (with only 45 patients treated).

Comments and Suggestions for Authors:

The manuscript is an interesting study, but requires some considerations.

Title:

The Title should appear with a question mark.

Abstract:

The type of study design should be indicated.

Line 19 and following. Why are the acronyms ERI used instead of RIF?

After reading the Abstract, the conclusions are not supported by the data presented in it.

2. Material and methods:

It is classified as an observational study on a longitudinal prospective cohort. However, the study is not observational, since it is therapeutically intervened.

It is not shown how the calculation of the sample size necessary to obtain conclusions about the stated objectives was carried out. This is important to present.

For the interleukin dysregulation interpretation, "the ratio is considered low when it is lower than the mean - 1SD of the control group and considered high when it is higher than the mean + 1 SD of the control group. Profiles of excess immune activation, or hypoimmune activation, or profiles within normal limits are defined according to the decision tree described previously [31]". However, the values ​​of + 1DS or - 1DS were not applied on all occasions in the cited study. Please clarify this issue.

A Statistical Analysis subsection is absolutely necessary.

3. Results:

Table 1 and 2. Decimals must be separated by a period and not by a comma.

Table 1 and 2. The variables should have been more specifically defined in the Material and Methods Section.

How do I consider "Parity (average) = 0.3"? Please clarify

“Average time biopsy / Pregnancy (months) 7 7 4”. Should be more strongly considered with media + DS

Line 212 and following. The results are presented in a very repetitive way as they appear in Table 2.

Line 232. It is indicated that “no significant difference between the groups was observed for all the parameters analyzed”. However, the statistical methods used are not indicated.

4. Discussion:

In general, little is disputed with the results of other similar studies.

The authors honestly acknowledge the limitation of the small number of samples to obtain results. They should also take into account the methodological deficits with which they are presented. The calculation of the necessary sample size or the statistical analyzes performed are not indicated.

This should be shown in the in the Abstract and Conclusions sections of the manuscript.

5. Conclusions

The conclusions are not supported by the results presented. If "no significant difference between the groups was observed for all the parameters analyzed", why is it concluded that "our study shows a clear improvement in the results of the rate of progressive pregnancies observed for our RIF patients after biopsy and management adapted to the endometrial immune profile"?

6. References:

References should be thoroughly revised to conform to uniform and appropriate standards for the journal Biomedicines.

Reviewer 2 Report

The study of the immune profile of the endometrium is important to a successful implantation. The topic is relevant and exciting to the field of the journal. The text is clear and easy to read. The manuscript has an excellent methodical description. The overall paper is organized and well-written. The methods, the overall study design, and the statistical analysis are clearly described. I appreciate that the results achieve the proposed objectives. The discussions section is well-organized, insightful, and informative. The figures and the tables are well-presented and easy to read and understand. The conclusions are clear and supported by the results. The references are up-to-date and appropriate for the theme.

Congratulations to the research team on this success.

Reviewer 3 Report

Coutanceau et al. they present a well-structured manuscript with interesting data. However, there are points that need to be improved:

-The authors include a title that is too broad and does not reflect the findings shown.

-The authors must improve the abstract, the authors must include a more adequate explanation of the results obtained.

-The introduction to the state of the art is adequate. However, authors should reduce the number of citations. Some references are too old and others are redundant.

-The authors should include in the spirit of the study a more translational perspective of the potential of this manuscript.

-The material and methods should be improved, the authors should explain this point in more detail. Authors must include specific references.

-An essential point is the sample size. The authors must justify the sample size with statistical methods.

-The authors must justify the exclusion and inclusion criteria.

-The authors must justify the statistical tests used.

-The authors must improve essential points of the presentation of the results. Figures should improve quality.

-The figure legends are very simple and do not contain enough information.

-The authors must improve the information in the tables, some data are incomplete as in table 1.

-The authors make an adequate discussion, but from line 296 to 304 they must do it in a narrative way and not point by point.

-The authors must justify and discuss the limitations of the study in an appropriate manner.

-The authors should improve the use of English grammar.

Round 2

Reviewer 1 Report

Round 2.

Authors make changes to the manuscript. Most of the corrections are summarized in the responses to the reviewer as "Done" without explaining where and how they were made. The extensive grammatical corrections made make it very difficult to make a new revision of the manuscript that is full of cancellations.

The methodological problem of the statistical tests used to detect significance persists. According to the authors "the Mann-Whitney test was used to compare groups", without explaining why this was done. This test is the non-parametric version of the usual Student's t-test for ordinal or continuous variables. How then did the proportions compare?

No statistical significance is found in the results presented to support the conclusions.

 The limitation accepted by the authors of the small number of cases persists and the fact that they did not calculate the necessary sample size for the proposed objectives.

Author Response

Firstly, my co-authors and I would like to thank you for giving us the opportunity to improve our manuscript. Thank you for your kind comments.

Reviewer 1:

Authors make changes to the manuscript. Most of the corrections are summarized in the responses to the reviewer as "Done" without explaining where and how they were made. The extensive grammatical corrections made make it very difficult to make a new revision of the manuscript that is full of cancellations.

We apologize for the inconvenience. The revisions (other than purely grammatical corrections) are now highlighted in yellow.

The methodological problem of the statistical tests used to detect significance persists. According to the authors "the Mann-Whitney test was used to compare groups", without explaining why this was done. This test is the non-parametric version of the usual Student's t-test for ordinal or continuous variables. How then did the proportions compare?

We apologize for not specifying how we compared proportions. We have revised the sentence, as follows: “To compare proportions, we used either the Mann-Whitney U test (for continuous variables) or Fisher’s exact test (for categorical variables).” Line 243.

Furthermore, we have added an explanatory column (specifying the statistical test used in each case) to the two tables. The column could be removed from the final version, if preferred.

No statistical significance is found in the results presented to support the conclusions.

We now refer to a “potential positive effect” in our conclusions in the abstract (line 40) and in the main text (line 388).

The limitation accepted by the authors of the small number of cases persists and the fact that they did not calculate the necessary sample size for the proposed objectives.

We have answered this question previously. Unfortunately, we did not calculate the required sample size. Our initial objective was to duplicate the (observational) study by Ledée et al. However, we agree that our study could be considered as being interventional. We now mention the lack of a sample size calculation as a study limitation.

Reviewer 2:

The authors have appropriately made the indicated changes.

However, the figures are still of low quality and authors must include a graphical summary.

In order to explain the decision tree, we have added a graphical summary before Figure 1. “To classify the patients, we sequentially used the IL18/TW ratio, the uNK cell count (when the IL18/TW ratio was normal), and the IL15/Fn14 ratio (when the uNK cell count was normal or high) Line 185. With regard to Figure 2, we have added the following sentence to the text: “The study’s patient flow is summarized in Figure 2”. Line 246.

Please, the authors should include translation more specifically in the discussion and better discuss the limitations of the study.

We have added the following comments at line 305: “However, our interventional, longitudinal, prospective cohort study with the same methodology as the previously reported study also had the same limitations. To validate the treatment strategy, a randomized, controlled trial would be required. ”.and line 372: “ Lastly, and as mentioned above, this was only an interventional, longitudinal, prospective cohort study that lacked a control group. However, it is very difficult not to suggest a new strategy for couples with RIF.”.

Grammatical errors are still numerous.

We apologize for having left half a dozen typos in the abstract. These have been corrected. We are confident that the main text was and still is grammatically and idiomatically correct. It has been copy-edited by a native English speaker (PhD in Biochemistry and a professional medical writer for the last 20 years, having drafted or copy-edited over 1100 manuscripts).

Reviewer 3 Report

The authors have appropriately made the indicated changes. However, the figures are still of low quality and authors must include a graphical summary. Please, the authors should include translation more specifically in the discussion and better discuss the limitations of the study.

Grammatical errors are still numerous.

Author Response

(The authors gave the same response as above.)

Round 3

Reviewer 1 Report

Round 3.

The authors state that "To compare proportions, we used either the Mann-Whitney U test (for continuous variables) or Fisher's exact test (for categorical variables)"? and that "We did not observe any significant differences between the three groups for any of the variables". However, they do not provide the p value found in the statistical tests.

Likewise, they do not indicate whether the data distributions were normal or not. By opting for the Mann-Whitney U test it seems that the data were non-parametric. However, as it was a comparison of 3 study groups, the test used should have been the Kruskal-Wallis test for non-parametric data and 3 groups.

Authors should incorporate statistical advice.

Author Response

Firstly, my co-authors and I would like to thank you again for giving us the opportunity to improve our manuscript. Thank you for your kind comments.

Reviewer 1:

The authors state that "To compare proportions, we used either the Mann-Whitney U test (for continuous variables) or Fisher's exact test (for categorical variables)"? and that "We did not observe any significant differences between the three groups for any of the variables". However, they do not provide the p value found in the statistical tests.

As stated in the text, if the p-value is greater than 0.05, it is considered insignificant. We have only added in the text (p>0.05) for clarity, and the p-value in the table as requested by the academic editor.

Likewise, they do not indicate whether the data distributions were normal or not. By opting for the Mann-Whitney U test it seems that the data were non-parametric. However, as it was a comparison of 3 study groups, the test used should have been the Kruskal-Wallis test for non-parametric data and 3 groups.

The Student's t test can be used if the number of data is greater than 30. This is not the case for 2 groups, and the Mann-Whitney test have been used when comparing the groups 2 to 2, considering that the hypo-activation group could have a significant impact on the result, as the number of patients was clearly different compared to the other 2 groups. This explains why we did not use the Kruskal-Wallis test. However, in accordance with the reviewers' and editor's remarks, we sought statistical advice and used the Kruskal-Wallis test. Regarding the distributions, they were normal for all continuous variable except for gravida, parity and attempt rank.

Authors should incorporate statistical advice.

Unfortunately, due to the short deadline (3 days) requested to submit a revised version by the editor and the Christmas period, it has been impossible for me to get a statistical advice in time. Therefore, and as previously written, we performed the statistical analysis as recommended.

Academic Editor

Authors should indicate whether the data distributions were normal or not. Authors should incorporate statistical advice. Because it was a comparison of 3 study groups, the test used should have been the Kruskal-Wallis test for non-parametric data and 3 groups. Furthermore, they should provide the p value found in the statistical tests in Table but not “MW” or “F”.

It has been done, except the statistical advice as previously written, due to the short deadline (3 days) requested to submit a revised version. If necessary, this could be done but not until early January.